# Effect of Antiparasitic Management of Cattle on the Diversity and Functional Structure of Dung Beetle (Coleoptera: Scarabaeidae) Assemblages in the Colombian Caribbean

Hernando L. Tovar [1], César M. A. Correa [2], Jean-Pierre Lumaret [3], Pablo A. López-Bedoya [4,5], Blas Navarro [6], Valentina Tovar [7] and Jorge Ari Noriega [8,*]

1   Institución Educativa Técnico Agropecuario El Piñal, Los Palmitos 701050, Colombia
2   Laboratory of Entomology, Universidade Estadual de Mato Grosso do Sul, Aquidauana 79200-000, Mato Grosso do Sul, Brazil
3   Laboratoire de Zoogéographie, Université Paul Valéry, Route de Mende, CEDEX 5, 34199 Montpellier, France
4   Programa de Pósgraduação em Entomologia, Departamento de Entomologia, Universidade Federal de Lavras, Lavras 37203-202, Minas Gerais, Brazil
5   Grupo de Investigación GEBIOME, Facultad de Ciencias Exactas y Naturales, Universidad de Caldas, Manizales 170004, Colombia
6   Institución Educativa Técnico Acuícola de Cascajal, Magangué 132518, Colombia
7   Institución Educativa Técnico Agropecuario, Universidad del Atlántico, Barranquilla 080007, Colombia
8   Grupo Agua, Salud y Ambiente, Facultad de Ingeniería, Universidad El Bosque, Bogotá 110121, Colombia
*   Correspondence: jnorieg@hotmail.com

**Abstract:** The transformation of forests into agricultural and livestock systems negatively affects the ecological dynamics and the ecosystem services provided by different groups of insects, including dung beetles, which stand out for their importance in recycling livestock dung. Since the 1980s, farmers in different regions of the world have been using Ivermectin to control parasites that affect cattle. The main route of elimination of this molecule and its metabolites is through manure, which affects the richness, abundance, and biomass of dung beetles when they use dung from treated animals. To quantify this effect, we carried out an experimental design in the field in the Colombian Caribbean, where nine cattle farms were evaluated, of which three were taken for each of the different cattle management practices most used in the region: (i) Ivermectin not applied, (ii) two doses of Ivermectin at 1% applied per year and (iii) two doses of Ivermectin at 3.15% applied per year. To assess the richness, abundance, biomass, and functional groups of dung beetles, during the dry and wet seasons, 30 pitfall traps were baited on each farm with fresh cattle manure with the same management doses described above. A total of 25,441 individuals belonging to 19 genera and 30 species were collected. The richness, abundance, and biomass of beetle assemblages decreased along the gradient represented by management without using Ivermectin and management where Ivermectin was used. Paracoprid beetles were the functional group that was most negatively affected in cattle farms with Ivermectin use. In cattle farms where Ivermectin was not used, there was a greater diversity and higher functional structure of dung beetle assemblages than in those where this veterinary medicinal product was used. Using Ivermectin generates short- and long-term effects on the richness, abundance, biomass, and functional groups of dung beetles in livestock systems in the Colombian Caribbean. Therefore, we suggest using integrated treatment management to prevent the recycling fauna from being affected.

**Keywords:** Aphodiinae; ecosystem services; grassland; Ivermectin; livestock management; Scarabaeinae; seasons

## 1. Introduction

With the development of livestock systems, the rate of conversion of forest cover areas into large areas of monocultures (pastures) for livestock feeding has increased; although

livestock production increased, biodiversity, habitat structure, and ecosystem services were negatively affected [1] Grasslands have a predominance of grasses that develop due to the interaction between climate, soil, and biota, and, often, these interactions are affected by anthropogenic activities, such as livestock overgrazing, the type of grazing, the use of agrochemicals for fertilization, insect control, pest control, and the use of veterinary medicinal products in livestock [2]. In addition, a process of soil degradation begins due to erosion, loss of organic matter, and surface compaction due to livestock trampling and agricultural mechanization.

According to the Colombian Federation of Livestock Farmers [3], cattle farming is the economic activity with the most significant presence in the Colombian countryside; it is found in all ecosystems and in different production systems, such as breeding, rearing, fattening, specialized dairy, and dual purposes. Livestock is the main agricultural activity in the country, and it generates approximately 810,000 direct jobs (6% of national employment) and contributes 1.4% of the country's gross domestic product. By 2022, the livestock inventory was 28.2 million heads, ranking 7th worldwide, with 512,000 cattle farms dedicated to the bovine activity, and 67.1% of them had less than 25 animals [4]. According to the Instituto Geográfico Agustín Codazzi [5], only 2.4% (2.7 million hectares) of Colombian land is suitable for livestock, but there are currently more than 14 million hectares occupied by livestock, which is over 11 million hectares with livestock land use conflicts.

One of the by-products of livestock farming in pastures is manure, which is the primary food source and nesting site for dung beetles (Coleoptera: Scarabaeidae) [6]. Within the degradation of excrement in the soil, two processes are carried out: (i) one is biotic and accompanied by a rich fauna that includes beetles, ants, termites, flies, and micro-organisms such as bacteria and fungi [7], (ii) and the other process is abiotic, where climatic factors such as temperature, rainfall, and soil moisture play a role [8]. Within the coprophagous fauna, dung beetles are essential elements for the ecosystem functioning in grasslands [9,10]. The adult dung beetles feed on the liquid suspension of the excrement, while the larvae eat the solid remains of the plants that were not digested by the cows [11]. Thus, these beetles, in the process of digging tunnels in the soil, improve soil aeration (i.e., macroporosity) and fertility [12,13], which accelerates microbial metabolism where compounds such as phosphorus, potassium, ammonia, nitrogen, and carbon present in the recycled manure are rapidly released [13,14]. In this process, the infiltration of water into the soil is improved [12], which facilitates grass growth [15], and the structure and water retention capacity of the soil is preserved [16,17]. Dung beetle activity thus prevents the long-term accumulation of manure in pastures [18]. In addition, dung beetles control fly populations [19] and reduce the re-infection of livestock by gastrointestinal nematodes that have developed in dung [20].

Sanitary management has become one of the most limiting factors in livestock production worldwide [21]. Consequently, veterinary drugs may affect the regular interaction of soil-degrading fauna such as dung beetles, dipterans, edaphic mesofauna, and earthworms [22,23]. Depending on the chemical family to which they belong, their mode of administration, and their residual concentration in excreted dung, the veterinary drugs vary in their toxicity to dung beetles [24]. From the 1980s onwards, the development of macrocyclic lactones made it possible to control both internal and external parasites of livestock (endectocides) [25,26]. From among these inexpensive and broad-spectrum parasiticides, Ivermectin has become the most widely used antiparasitic by livestock farmers [27]. Ivermectin is administered in different forms, which include topically (pour-on), orally, or by subcutaneous injection. In all cases, the molecule is not completely metabolized by the animals, and the parent molecule remains intact in the feces for a long time, often for several months [28].

Once Ivermectin is applied to cattle, its residues begin to appear in the feces from the first days [29], and toxic dung affects dung beetles by impacting their oviposition, fertility [30,31], decreasing their olfactory and locomotor capacity [32], and modifying their attraction to dung [33,34]. Even low doses of Ivermectin can significantly affect

them [32]. In addition, the regular use of Ivermectin in cattle leads to a significant reduction in the abundance and richness of beetles, which slows the degradation of manure in pastures [18,30,34,35] and disturbs the soil nutrient cycle [14,18]. The physiological and behavioral disorders produced by Ivermectin in the short-term in dung beetles may have a long-term impact on the structure of their assemblages and their functional efficiency at the ecosystem level [9,18]. Although studies on the unintended effects of Ivermectin on dung beetle assemblages have been conducted recently worldwide, such as in Europe [36], Canada, the United States [34,37,38], and Mexico [39–41], in South America these studies are still scarce (only in Colombia: [42] and Brazil: [35]).

The present study aims to assess the unintended effects of Ivermectin use on the richness, abundance, and biomass of dung beetles in Colombian Caribbean cattle farms by comparing farms where Ivermectin is not used or is used in different doses (i.e., 1% and 3.15%), during distinct periods of the year (dry and rainy seasons). The objective was to answer the following questions: (i) What is the effect of the use of Ivermectin on dung beetle assemblages (richness, abundance, and biomass) in cattle farms in the Colombian Caribbean? (ii) Do the functional groups of dung beetles present different responses for use of Ivermectin? (iii) Are there species associated with each use of Ivermectin (e.g., dung without IVM, with a low dose (IVM 1%), and with a high dose (IVM 3.15%)? As a hypothesis, we expect that the use of Ivermectin in Colombian Caribbean livestock production systems will generate adverse effects on the structure (composition, richness, abundance, functional groups, and biomass) of the dung beetle assemblages.

## 2. Material and Methods

### 2.1. Study Area

The study area covers three different municipalities: Los Palmitos, Morroa, and Sincelejo in the department of Sucre, Caribbean region of Colombia (Figure 1). The temperature in this region ranges between 24 and 32 °C, and rainfall varies between 858 and 1607 mm annually on average [43]. The rainy season runs from April to November, and the dry season runs from December to March. Due to its climatic characteristics, the region corresponds to the Dry Forest (Bs-T) life zone [44]. Livestock in this region is mainly made up of breeds of *Bos primigenius* L., 1758 (zebu cattle). Cattle are usually managed in extensively used pastures to produce milk and meat (dual-purpose farming), and animals are rotated in two or three paddocks. The selected cattle farms present a silvopastoral configuration, where grasses abound (*Bothriochloa pertusa* (L.) A. Camus, F. Poaceae) in association with trees and shrubs; the surface area of each farm is approximately 30 ha, and the cattle graze all year round. The average stocking rate is 1.5 animal units per hectare (1.5 AU/ha), considering one unit as 450 kg of average live weight.

### 2.2. Livestock Management

To assess and compare the effect of the use of Ivermectin (IVM) on the diversity and structure of dung beetle assemblages, the study was carried out on nine livestock farms in the Colombian Caribbean, at least 3 to 5 km apart from each other. Selected farms represent the typical parasite control in the Colombian Caribbean region: (i) NO-IVM = Ivermectin was not applied (nor any other product for more than 30 years), (ii) TWO-IVM 1% = Two doses of Ivermectin at 1% concentration per year, and (iii) TWO-IVM 3.15% = Two doses of Ivermectin at 3.15% concentration per year. These two doses were usually applied in February (dry season) and in October (rainy season).

### 2.3. Sampling and Experimental Design

To collect dung beetles, pitfall traps baited with fresh manure from the cattle of each farm under study were used, where three groups of seven cattle were taken for each of the treatments used. The first group remained without treatment (control group), the second was given IVM at 1% (low dose Ivermectin; trade name Ivomec®), and the third group was given IVM at 3.15% (high dose Ivermectin; trade name Ivomec® Gold). One ml of IVM was

used subcutaneously in the neck region for every 50 kg of body weight (each dose provides 200 μg and 630 μg of Ivermectin (1% and 3.15%) per kg of body weight, respectively). Using dung from treated and not controlled animals as baits helped determine whether the beetles were attracted to any of them.

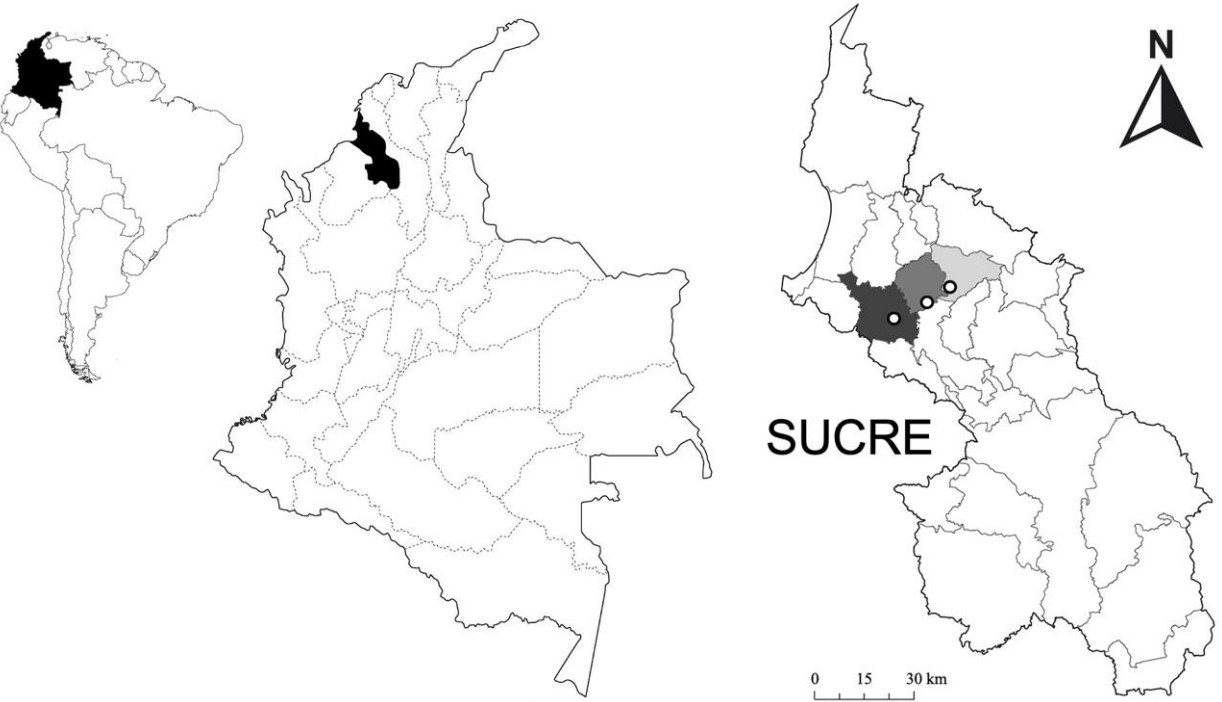

**Figure 1.** Location of the study sites in the municipalities of Sincelejo (black), Morroa (dark gray), and Los Palmitos (light gray), Sucre—Colombia.

As each season is characterized by distinct assemblages of dung beetles [45,46], the fieldwork was carried out in the selected cattle farms during the dry season (February) and the rainy season (September) of 2020. The experimental design included three factors: (i) livestock management related to the use of Ivermectin: NO-IVM, TWO-IVM 1%, and TWO-IVM 3.15%; (ii) season: dry and rainy; and (iii) treatments related to the presence of Ivermectin in the baits: without IVM, IVM 1 %, and IVM 3.15%. The pitfall traps consisted of a 3.5 L plastic container with a circular opening of 20 cm in diameter and a height of 15 cm, which was buried at ground level and filled with approximately 500 mL of a preservative solution (50% of water, 50% alcohol at 70%, and a few drops of neutral liquid detergent). A plastic net with a 5 cm diameter opening was placed over the trap, holding the bait (about 1000 g), enclosed in a gauze cloth.

In each farm, 30 sampling traps were installed [47], which were placed in the paddocks in grids 50 m apart [48] and were reviewed and collected after 48 h [40]. Of the 30 baits used in the traps, 10 had excrement without IVM, 10 had a low dose (IVM 1%), and 10 had a high dose (IVM 3.15%). The treatments were randomly located in the paddocks. The manure for the baits was obtained from the cattle treated on the study farms four days after the antiparasitic product was applied, which is the time necessary to ensure that the substance was present in the dung [49]. The manure was collected in the morning before beginning the activity of the beetles to avoid colonization [32]. A total of 540 pitfall traps (270 traps for each season, rainy and dry) were placed in the field to determine variations in the diversity and structure of the dung beetle assemblages.

### 2.4. Dung Beetles' Diversity, Identification, and Biomass

For each pitfall trap, the number of species (richness), the number of individuals per species (abundance), and the biomass per species and total (g) were counted. Captured

beetles were identified to the species level using different taxonomic identification keys for Scarabaeinae [50–55] and Aphodiinae [56–59]. Specimens were deposited in the senior last author's reference collection (CJAN). Biomass per species was quantified using a precision balance (Ohaus Pioneer PA323 0.001 g, Ohaus Corporation, Sunnyvale, CA, USA).

*2.5. Data Analysis*

A list of the species collected in the samplings was compiled, which were grouped according to two criteria. The first took into account the way in which beetles use and relocate dung (paracoprids, telecoprids, and endocoprids) [60], and the second was recorded according to body size, with species divided into small (<10 mm), medium (10 to 18 mm), and large (>18 mm) [61], thus establishing nine functional groups [6]: small paracoprids (Pp), small telecoprids (Tp), small endocoprids (Ep), medium paracoprids (Pm), medium telecoprids (Tm), medium endocoprids (Em), large paracoprids (Pg), large telecoprids (Tg), and large endocoprids (Eg). Then, we estimated the sample completeness for each livestock management using a sample coverage analysis with iNEXT package (see [62,63]) in R version 3.2.1 [64]. In addition, we plotted species rank abundance distributions to visually compare patterns of species dominance in the dung beetle assemblage sampled in each livestock management type.

We used Generalized Linear Mixed Models (GLMMs) to verify the effect of different livestock management approaches (explanatory variable) on the abundance, species richness, and biomass (response variables) of dung beetles, with livestock management as a fixed factor and sampling season (dry and rainy) as a random factor. We classified the sampled species into three functional groups related to their nesting behavior: endocoprids (dwellers), telecoprids (rollers), and paracoprids (tunnelers) [60]. In addition to analyzing the total abundance, species richness, and biomass data, we also used GLMMs to analyze the effects of livestock management on the abundance, species richness, and biomass of each dung beetle functional group separately. We used the Poisson error for species richness, which is a negative binomial error distribution with log link function for abundance, as these data showed over-dispersion, and we used the Gaussian error for biomass [65]. These analyses were undertaken using the "glmer.nb" function in the LME4 package in R version 3.2.1 [64]. We used the indicator value method following Dufrêne and Legendre [66], to identify associations between dung beetle species and types of bait (see [67]). We used 5000 randomizations to determine the statistical significance of the observed indicator value (Monte Carlo test; $p < 0.05$). This analysis was performed with the 'indicspecies' package in R software [64,68].

**3. Results**

*3.1. Seasonal Differences in Dung Beetle Assemblages*

A total of 25,441 individuals of the Scarabaeidae family belonging to the subfamilies Aphodiinae and Scarabaeinae were collected and grouped into 30 species, 19 genera, and nine tribes (Table 1). The Scarabaeinae subfamily was represented by 25 species (n = 10,617, 41.7%), followed by Aphodiinae with five species (n = 14,824, 58.3%) (Table 1). The genera with the highest number of species were *Canthon* (n = 5), *Onthophagus* (n = 4), *Ataenius*, *Uroxys*, *Dichotomius*, and *Eurysternus* (n = 2; Table 1). For the first time, 10 new species, identified at the species-specific level, were reported for this region according to the list by Noriega et al., (2013): *Ataenius complicatus* Harold, 1869, *Ataenius crenulatus* Schmidt, 1910, *Cartwrightia cartwrighti* Cartwright, 1967, *Coprophanaeus gamezi* Arnaud, 2002, *Dichotomius coenosus* (Erichson, 1848), *Nialaphodius nigrita* (Fabricius, 1801), *Onthophagus buculus* Mannerheim, 1829, *Onthophagus clypeatus* Blanchard, 1846, *Sylvicanthon aequinoctialis* (Harold, 1868), and *Xenoheptaulacus tricostatus* (Harold, 1869). *Cartwrightia* is a new genus record for Colombia [69]. It is interesting to note the presence of two introduced species in the region: *Digitonthophagus gazella* (Fabricius, 1787) and *N. nigrita*. The sample coverage estimator revealed a high sampling efficiency (>99% in all livestock management) (Table 1).

This indicates that we had conducted an adequate effort to represent the dung beetle assemblages in our sampling sites.

**Table 1.** List of dung beetle species and number of individuals collected in the nine cattle farms of the Caribbean region (Colombia). Livestock management: No-IVM = Ivermectin is not applied per year; Two-IVM 1% = Two doses of Ivermectin 1% applied per year; and Two-IVM 3.15% = Two doses of Ivermectin 3.15% applied per year. Functional groups: small paracoprids (Pp), small telecoprids (Tp), small endocoprids (Ep), medium paracoprids (Pm), medium telecoprids (Tm), medium endocoprids (Em), large paracoprids (Pg), large telecoprids (Tg), and large endocoprids (Eg). Season (Ll = Rainy and S = Dry).

| SubFam/ Tribe | Species | FG | No-IVM | | Two-IVM 1% | | Two-IVM 3.15% | | Total (%) |
|---|---|---|---|---|---|---|---|---|---|
| | | | Ll | S | Ll | S | Ll | S | |
| Aphodiinae | | | | | | | | | |
| Aphodiini | *Nialaphodius nigrita* (Fabricius, 1801) | Ep | 55 | 334 | 6 | 1241 | 4 | 177 | 1817 (7.1) |
| | *Xenoheptaulacus tricostatus* (Harold, 1869) | Ep | 2 | 0 | 0 | 0 | 0 | 0 | 2 (0.0) |
| Eupariini | *Ataenius crenulatus* Schmidt, 1910 | Ep | 6235 | 160 | 4755 | 43 | 1712 | 15 | 12,920 (50.8) |
| | *Ataenius complicatus* Harold, 1869 | Ep | 57 | 2 | 7 | 0 | 15 | 0 | 81 (0.3) |
| | *Cartwrightia cartwrighti* Cartwright, 1967 | Ep | 2 | 1 | 1 | 0 | 0 | 0 | 4 (0.0) |
| Scarabaeinae | | | | | | | | | |
| Ateuchini | *Ateuchus* sp. 1 | Pp | 2 | 0 | 0 | 0 | 1 | 0 | 3 (0.0) |
| | *Uroxys* sp. 1 | Pp | 0 | 3 | 0 | 0 | 0 | 0 | 3 (0.0) |
| | *Uroxys* sp. 2 | Pp | 1 | 0 | 0 | 1 | 0 | 0 | 2 (0.0) |
| Coprini | *Canthidium* sp. 1 | Pp | 0 | 0 | 0 | 0 | 1 | 0 | 1 (0.0) |
| | *Dichotomius agenor* (Harold, 1869) | Pg | 84 | 0 | 62 | 0 | 10 | 0 | 156 (0.6) |
| | *Dichotomius coenosus* (Erichson, 1848) | Pg | 2 | 0 | 0 | 0 | 0 | 0 | 2 (0.0) |
| Deltochilini | *Canthon juvencus* Harold, 1868 | Tp | 0 | 0 | 0 | 1 | 0 | 0 | 1 (0.0) |
| | *Canthon lituratus* (Germar, 1813) | Tp | 33 | 0 | 2 | 0 | 15 | 0 | 50 (0.2) |
| | *Canthon mutabilis* Lucas, 1859 | Tp | 43 | 1 | 10 | 0 | 75 | 0 | 129 (0.5) |
| | *Canthon septemmaculatus* (Latreille, 1812) | Tm | 9 | 0 | 0 | 0 | 64 | 0 | 73 (0.3) |
| | *Canthon* sp. 1 | Tp | 0 | 0 | 0 | 0 | 1 | 0 | 1 (0.0) |
| | *Deltochilum guildingii* (Westwood, 1835) | Tg | 1 | 0 | 0 | 0 | 0 | 0 | 1 (0.0) |
| | *Pseudocanthon xanthurus* (Blanchard, 1846) | Tp | 1 | 0 | 0 | 0 | 0 | 0 | 1 (0.0) |
| | *Sylvicanthon aequinoctialis* (Harold, 1868) | Tp | 1 | 0 | 0 | 0 | 0 | 0 | 1 (0.0) |
| Demarziellini | *Trichillidium pilosum* (Robinson, 1948) | Pp | 1 | 0 | 0 | 0 | 0 | 0 | 1 (0.0) |
| Oniticellini | *Eurysternus impressicollis* Castelnau, 1840 | Ep | 188 | 12 | 34 | 8 | 48 | 1 | 291 (1.1) |
| | *Eurysternus mexicanus* Harold, 1869 | Em | 67 | 4 | 11 | 2 | 13 | 1 | 98 (0.4) |
| Onthophagini | *Digitonthophagus gazella* (Fabricius, 1787) | Pm | 1453 | 54 | 808 | 30 | 502 | 18 | 2865 (11.3) |
| | *Onthophagus bidentatus* Drapiez, 1819 | Pp | 2 | 0 | 0 | 2 | 4 | 0 | 8 (0.0) |
| | *Onthophagus buculus* Mannerheim, 1829 | Pp | 39 | 0 | 2 | 0 | 12 | 0 | 53 (0.2) |
| | *Onthophagus clypeatus* Blanchard, 1846 | Pp | 4 | 4 | 1 | 2 | 0 | 0 | 11 (0.0) |
| | *Onthophagus marginicollis* Harold, 1880 | Pp | 2469 | 6 | 3192 | 0 | 1188 | 0 | 6855 (26.9) |
| Phanaeini | *Coprophanaeus gamezi* Arnaud, 2002 | Pg | 1 | 0 | 0 | 0 | 0 | 0 | 1 (0.0) |
| | *Diabroctis cadmus* (Harold, 1868) | Pg | 1 | 0 | 0 | 0 | 8 | 0 | 9 (0.0) |
| | *Phanaeus hermes* Harold, 1868 | Pm | 0 | 0 | 1 | 0 | 0 | 0 | 1 (0.0) |
| | Abundance | | 10,753 | 581 | 8892 | 1330 | 3673 | 212 | 25,441 |
| | Richness | | 25 | 11 | 14 | 9 | 17 | 5 | 30 |
| | Biomass (g) | | 364.32 | 10.14 | 253.26 | 9.63 | 156.4 | 3.24 | 797.22 |

The rainy season registered a higher richness, abundance, and biomass of dung beetles (28 spp.; n = 23,318, 91.7%; 774.2 g, 97.1%). The most abundant species were *A. crenulatus* (n = 12,702, 49.9%), followed by *Onthophagus marginicollis* Harold, 1880 (n = 6849, 26.9%) and *D. gazella* (n = 2763, 10.9%) (Table 1). The paracoprids were the guild with the highest richness (14 spp.; n = 9851, 38.7%), followed by the endocoprids (7 spp.; n = 13,212, 51.9%),

and finally the telecoprids (7 spp.; n = 255, 1.0%) (Table 1). The dry season registered a lower richness, abundance, and biomass of beetles (14 spp.; n = 2123, 8.3%; 23.0 g, 2.9%), and the most abundant species was *N. nigrita* (n = 1752, 6.9%), followed by *A. crenulatus* (n = 218, 0.9%) and *D. gazella* (n = 102, 0.4%) (Table 1). The paracoprids (6 spp.; n = 120, 0.5%) and the endocoprids (6 spp.; n = 2001, 7.9%) were the richest guilds, followed by the telecoprids (2 spp.; n = 2) (Table 1). In terms of functional groups, no large endocoprids were found in either of the two seasons and for the dry season, and large paracoprids and telecoprids were absent, as well as medium telecoprids. For both seasons, small beetles made up the dominant group of the assemblage (20 spp., n = 22235, 87.4%) (Table 1). Of the 30 species recorded, 12 species were continuously present during the two seasons, with the most abundant being *A. crenulatus* (n = 12920, 50.8%), followed by *O. marginicollis* (n = 6855, 26.9%) and *D. gazella* (n = 2865, 11.3%) (Table 1); a total of 16 exclusive species were present during the rainy season and 2 species were present for the dry season (Table 1).

### 3.2. Differences in Dung Beetle Assemblages between Livestock Treatments

The NO-IVM treatment presented the highest richness, abundance, and biomass of dung beetles (26 spp.; n = 11,334, 44.6%; 374.5 g, 47.0%), followed by Two-IVM 1% (17 spp.; n = 10,222, 40.2%; 262.9 g, 33.0%), and, finally, TWO-IVM 3.15% (17 spp.; n = 3885, 15.3%; 159.9 g, 20.1%) (Table 1). NO-IVM and TWO-IVM 1% treatments were the most similar (81.6%), followed by TWO-IVM 3.15% treatment, which was the most different among all the treatments (58.4%). The IVM 3.15% and NO-IVM treatments were more similar (94.0%), followed by the IVM 1% treatment, which was more different (84.9%). The identity of the dominant species changed over the cattle removal age.

The identity of the dominant species changed over the cattle removal age. The identity of the dominant species changed among different treatments. However, *A. crenulatus*, *O. marginicollis*, and *D. gazella* were present among the three dominant species in the three treatments (Figure 2). Twelve species were recorded in all three treatments. The No-IVM had nine species recorded exclusively in this management, and it shared three species with TWO-IVM-1% and three species with TWO-IVM-3.15%. Two species were recorded exclusively in the TWO-IVM 1%; these treatments shared no species with the IVM 3.15%. Two species were exclusively collected in the TWO-IVM 3.15% treatment (Figure 3). The number of individuals was significantly lower in the TWO-IVM 3.15% than in the NO-IVM and TWO-IVM 1% treatment ($\chi^2_{2,4}$ = 10.35, *p* < 0.01; Figure 4A). The species richness was significantly higher in the NO-IVM than in the TWO-IVM 1% and TWO-IVM 3.15% ($\chi^2_{2,4}$ = 10.79, *p* < 0.01; Figure 4B). Regarding dung beetle biomass, the NO-IVM had higher biomass, followed by the TWO-IVM 1%, while the TWO-IVM 3.15% the treatment type had the lowest biomass (F$_{2,4}$ = 6.79, *p* = 0.03; Figure 4C).

### 3.3. Differences in Dung Beetle Relocation Guilds between Livestock Treatments

The number of individuals of endocoprids ($\chi^2_{2,4}$ = 9.62, *p* < 0.01) and paracoprids ($\chi^2_{2,4}$ = 5.74, *p* = 0.05) were significantly higher in the NO-IVM. However, the number of individuals of telecoprids ($\chi^2_{2,4}$ = 2.21, *p* = 0.33) did not differ among treatments (Figure 5A–C). The species richness of the paracoprids ($\chi^2_{2,4}$ = 10.79, *p* < 0.01) was significantly higher in the NO-IVM than in the TWO-IVM 1% and TWO-IVM 3.15%, but no differences were found for any of the other functional groups: endocoprids ($\chi^2_{2,4}$ = 2.84, *p* = 0.24) and telecoprids ($\chi^2_{2,4}$ = 4.85, *p* = 0.08) (Figure 5D–F). Finally, the biomass of the paracoprids (F$_{2,4}$ = 7.08, *p* = 0.03) was significantly higher in the NO-IVM than in the TWO-IVM 1% and TWO-IVM 3.15%. However, the biomass of the endocoprids (F$_{2,4}$ = 3.56, *p* = 0.16) and telecoprids (F$_{2,4}$ = 2.21, *p* = 0.33) did not significantly differ among treatments (Figure 5G–I). Of the 30 species evaluated for bait type preference, no one showed a preference for any bait type.

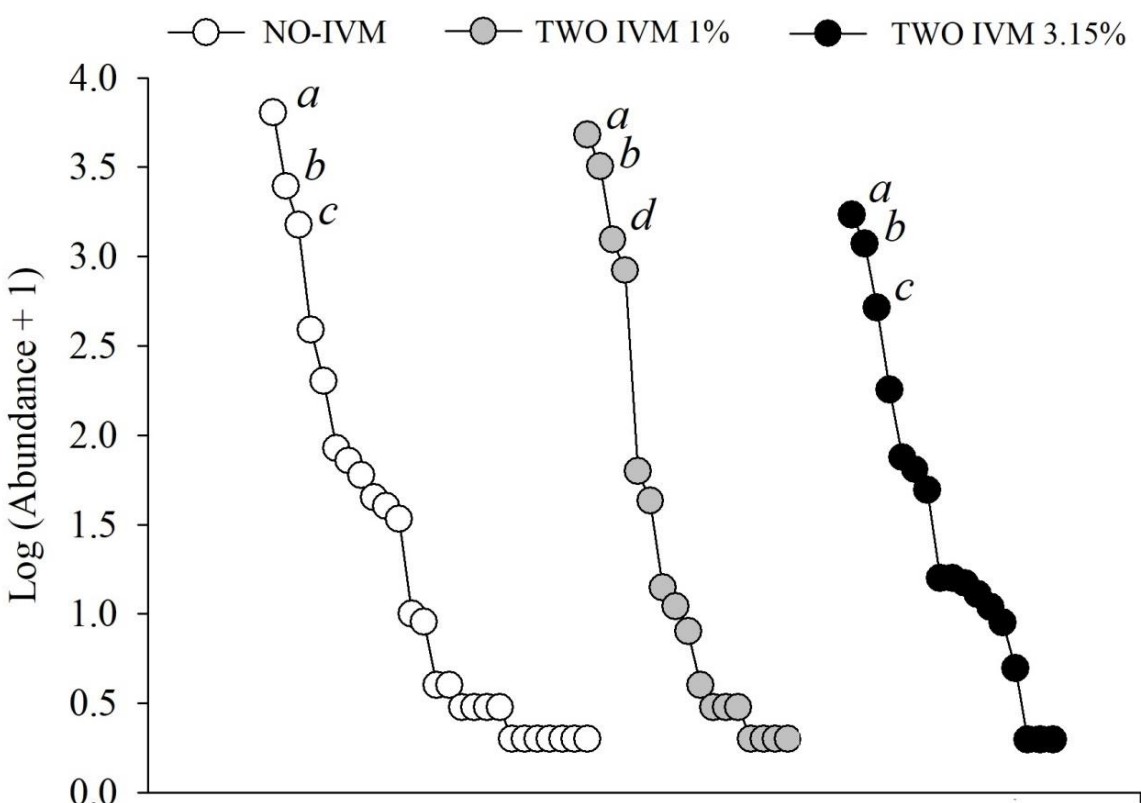

**Figure 2.** Rank distribution of dung beetle species across cattle farms with different livestock management in Colombian Caribean. *a*, *Ataenius crenulatus*; *b*, *Onthophagus marginicollis*; *c*, *Digitonthophagus gazella*; *d*, *Nialaphodius nigrita*.

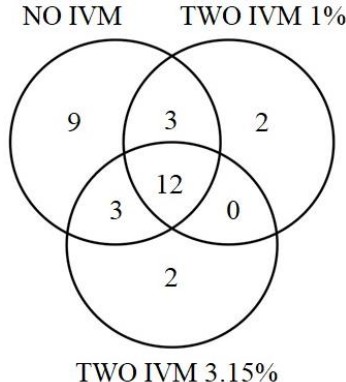

**Figure 3.** Venn diagram showing proportion of shared dung beetle species among the assemblages sampled in cattle farms with different Ivermectin treatments. Diagram components represent species unique to NO-IVM (circle left), the number of species unique to TWO-IVM 1% (circle right) and to TWO-IVM 3.15% (circle below), and species shared by the different livestock management approaches (overlap).

There was also a decreasing structure of beetle guilds in richness, abundance, and biomass, with a similar pattern in the NO-IVM (Paracoprids, 13 spp., n = 4126, 331.8 g > Endocoprids, 7 spp., n = 7119, 37.3 g > Telecoprids, 6 spp., n = 89, 5.4 g), in the most intervened TWO-IVM 1% (Paracoprids, 8 spp., n = 4101, 239.9 g > Endocoprids, 6 spp., n = 6108, 22.6 g > Telecoprids, 3 spp., n = 13, 0.4 g) and in the TWO-IVM 3.15% (Paracoprids, 8 spp., n = 1744, 136.9 g > Endocoprids, 5 spp., n = 1986, 9.6 g > Telecoprids, 4 spp., n = 155, 13.3 g) (Table 2). There was a tendency to lose species and complete functional groups as treatments without Ivermectin (NO-IVM) were passed towards the most intervened

treatments (TWO-IVM 1% and TWO-IVM 3.15%) (Table 2). A very significant loss of species was observed between farms where Ivermectin was not used (NO-IVM) compared to farms where IVM was used: 11 species were lost between NO-IVM and TWO-IVM 1% (Table 1; Figure 6); 11 species were also lost between the group of farms without IVM and the group of farms with TWO-IVM 3.15 % (Table 1 and Figure 6). A total of 14 species were lost between the control farms without Ivermectin and the management with Ivermectin application.

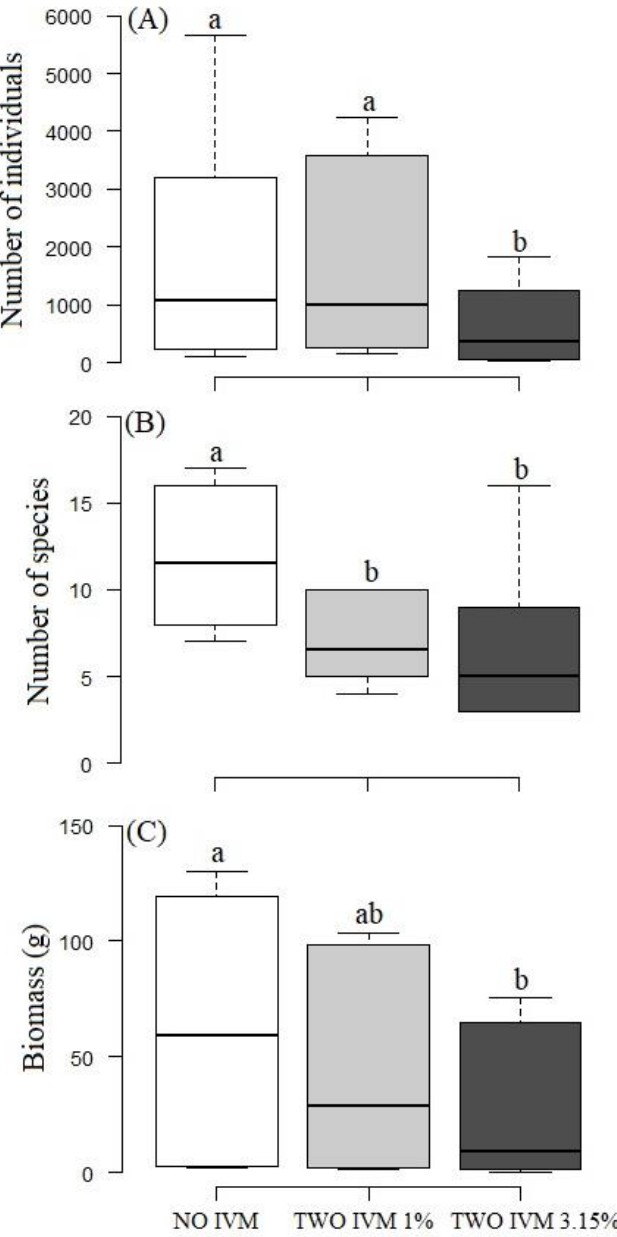

**Figure 4.** Boxplots of the average abundance (**A**), species richness (**B**), and biomass (**C**) of dung beetles sampled in cattle farms with NO-IVM, TWO-IVM 1%, and TWO-IVM 3.15% livestock management approaches in the Colombian Caribbean. Different letters above the boxes indicate statistically significant differences ($p < 0.05$).

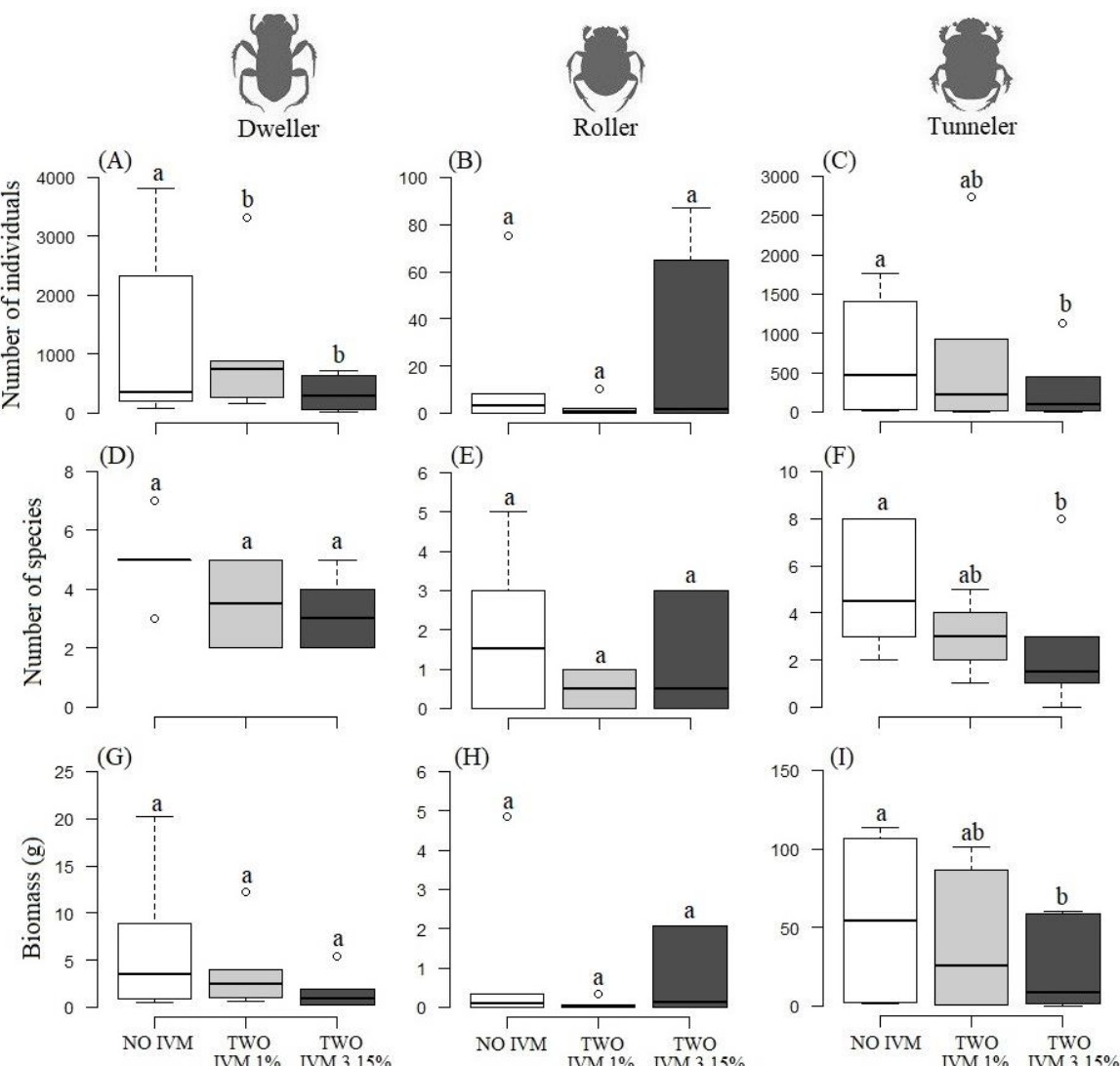

**Figure 5.** Boxplots of the average abundance (**A–C**), species richness (**D–F**), and biomass (**G–I**) of dung beetle functional groups sampled in cattle farms with NO-IVM, TWO-IVM 1%, and TWO-IVM 3.15% livestock management approaches in the Colombian Caribbean. Different letters above the boxes indicate statistically significant differences ($p < 0.05$).

**Table 2.** Richness, abundance, and biomass of the guilds (paracoprid, endocoprid, and telecoprid) of coprophagous beetles collected in nine livestock farms in the Caribbean region (Colombia) in 2020, with information on livestock management approaches on the farms (No-IVM = No Ivermectin is applied per year; Two-IVM 1% = Two doses of Ivermectin 1% applied per year; Two-IVM 3.15% = Two doses of Ivermectin 3.15% applied per year), type of relocation (TR; P = Paracoprids, T = Telecoprids, and E = Endocoprids).

| | Richness | | | Abundance | | | Biomass (g) | | |
|---|---|---|---|---|---|---|---|---|---|
| **Livestock Management** | | | | | **Guild** | | | | |
| | **P** | **E** | **T** | **P** | **E** | **T** | **P** | **E** | **T** |
| No-IVM | 13 | 7 | 6 | 4126 | 7119 | 89 | 331.8 | 37.3 | 5.4 |
| Two-IVM 1% | 8 | 6 | 3 | 4101 | 6108 | 13 | 239.9 | 22.6 | 0.4 |
| Two-IVM 3.15% | 8 | 5 | 4 | 1744 | 1986 | 155 | 136.9 | 9.6 | 13.3 |

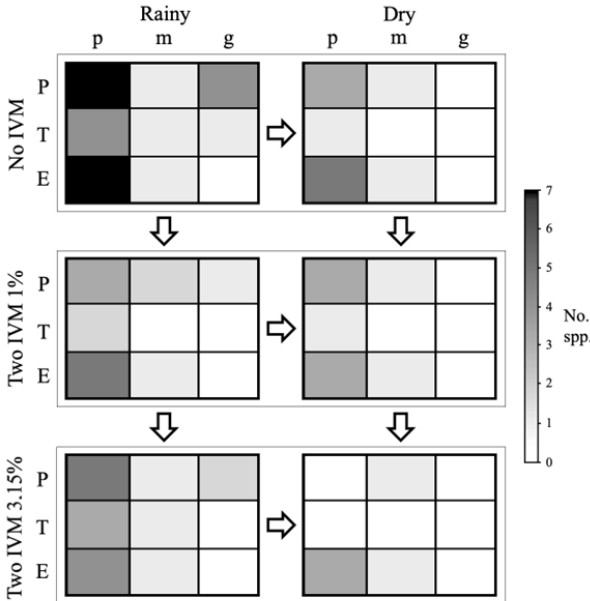

**Figure 6.** Structure of functional groups (Guilds: P = Paracoprids; T = Telecoprids, and E = Endoco-prids) and sizes of individuals (small, *p* < 10 mm; medium, m = 10 to 18 mm, and large, g > 18 mm), for livestock management (NO-IVM = Ivermectin is not applied per year, TWO-IVM 1% = Two doses of Ivermectin 1% applied per year, and TWO-IVM 3.15% = Two doses of Ivermectin 3.15% applied per year) in nine cattle farms in the municipalities of Los Palmitos, Sincelejo, and Morroa, department of Sucre (Colombia). The arrows indicate the possible directions in which the loss of groups can occur when increasing the concentrations or number of doses of Ivermectin.

## 4. Discussion

This study was carried out in nine cattle farms in the department of Sucre in the Caribbean region in a Bs-T life zone, which is an ecosystem with high diversity [70] and that is of importance in the provision of ecosystem services [71]. In the Neotropics, Bs-T is one of the most threatened habitats experiencing high transformation rates due to human activities [72,73]. Their transformation is mainly due to its proximity to large urban settlements that generate large agricultural and livestock areas [72,74]. By eliminating the forest, the vegetation cover is modified, and the land use changes, which modifies the environmental variables, producing changes in the composition and structure of the communities [75], which affects the functioning of ecosystems and the supply of services [76,77].

### 4.1. Dung Beetles' Assemblage Analysis in Tropical Dry Forests

This study, although carried out in a transformed Bs-T area (livestock farms), recorded the highest richness of dung beetles for the department of Sucre (30 species and 19 genera), which represent 46.2% of the known species in the region (n = 65 species). Although most of the studies carried out in this region have been carried out in forest areas, those areas showed a lower richness of dung beetle species (e.g., [42,45,46,78–82]). If we compare the dung beetle richness in the present study with the results recorded for the rest of the Caribbean region of Colombia in the Bs-T, we find that the only values that exceed it are those of Solis et al. [83] and Rangel-Acosta and Martínez [84]; the other studies giving a lower richness (e.g., [61,85–88].

The higher species richness recorded here is partly due to the inclusion of cattle farms that were adjacent to forest fragments, which contributed to maintaining essential key parts of the beetle diversity. Some configurations of agricultural and livestock systems have been found to help maintain biodiversity and ecosystem services [89]. This was demonstrated by Gascon et al. [90] and Mendenhall et al. [91], who found that, within large areas of agricultural and livestock use where forest remnants, live fences, thickets, fruit, and timber trees are preserved, this landscape structure plays a positive role through elements

contributing to maintaining species richness in the transformed matrix. In tropical regions, where local extinction rates are high due to habitat loss or fragmentation, secondary forests may, fortunately, support many forest species, while adjacent grasslands and clearings generally support fewer species, including few forest species [92]. These open areas in the forest matrix are mainly used by only a few generalist coprophagous species. Logging creates an extensive network of pathways within the forest, whose direct influence extends far beyond the pathway boundary [93]. Larger species, particularly paracoprids, are most affected and respond more than other functional groups of coprophagous beetles, but other groups are also influenced by changes in their microhabitat.

The genera with the highest number of species were *Canthon* and *Onthophagus*, and these results were similar to other studies from the Caribbean region of Colombia (e.g., [45,46,78–80,83,85,86,94,95]). The most abundant species in this ecosystem was *A. crenulatus* (Aphodiinae; see [96]), with more than 50% of the total individuals captured; followed by *O. marginicollis* and *D. gazella* which had been previously reported for the Bs-T in the department of Sucre (e.g., [45,78,80,82]). Despite not being an abundant species, it is important to highlight the presence of *D. guildingii*, as it is associated with well-preserved Bs-T ecosystems and has been proposed as a bioindicator for monitoring plans in the Caribbean region [72].

### 4.2. Dung Beetles' Assemblage Structure according to Seasonality

The greatest richness, abundance, and biomass of dung beetles occurred in the rainy season, since there is a high degree of association with fresh resources, which is evidenced by a greater distribution, composition, and structure of the assemblages [97]. These changes occur in the tropical region according to pluviometric regimes (e.g., [98–101]) and the dominant seasonal pattern in the Neotropics (e.g., [83,85,95,97,102,103]) responding directly to precipitation [104]. With the arrival of the rainy season, the soils increase their water content, which moistens the subterranean pupation chambers of dung beetles and facilitates the emergence of the new generations [94], while the rains favor the development of new leaves, flowers, and seeds in the pastures. The increase in plant biomass provides a better availability of food resources to livestock [101,105], which in turn allows the production of more and better-quality dung used for feeding and nesting by dung beetles [106].

In the dry season, the richness, abundance, and biomass of dung beetles were lower (i.e., 14 spp. species less than during the rainy season), which reduces the functional activity of the species, which is restricted to the peaks of humidity and temperature that condition their physiology [107]. In the dry season, soils are dehydrated due to the lack of rain, which limits the activity of beetles, who dig a few nests and shallow galleries because of the hardness and compaction of soils [98]. In the specific case of areas used for livestock production, their soils are often compacted by the overload of cattle and further suffer from erosion due to trampling, high insolation, and loss of humidity due to low tree cover; all this makes the soils hard and difficult to dig to build galleries and incorporate manure into the soil [106,108]. Finally, the quality of the dung is poor for dung beetles, as livestock often ingest hard, fibrous, and low-protein plants. This results in a reduction in the assemblage structures, which is probably related to the breeding season of the Scarabaeinae species [109]. For this type of ecosystem, the activity of adult beetles is seasonal, as they present larval or diapausal stages during the dry season [60], as in the case of several Aphodiinae species in Mexico [110]. Soil temperature is another critical variable related to increasing or decreasing the richness and abundance of beetles; when the soil temperature reaches 45 °C, it can kill larvae in open areas such as cattle pastures [111]. Some studies (see [112]) show that human deforestation interventions for agriculture and livestock production have favored the more heliophilic coprophagous species to the detriment of the more forest-dwelling species and species dependent on the dung of large herbivores, which agrees with our results. In the humid tropics, the species typical of anthropized habitats are mostly opportunistic, while more specialized species tend to seek forest habitats for feeding and breeding [48].

In this study, 12 generalist species were present in the two seasons and represented more than 98% of the captured individuals, of which ten were small, and only two were medium-sized beetles. This pattern shows that large beetles were the most affected species and were probably the first to disappear in such affected ecosystems, as has already been demonstrated for other tropical forest areas [93]. In contrast, the elevated abundances of the introduced species (*D. gazella* and *N. nigrita*) during the dry season drew attention. It is possible that they are less competitive species that are displaced to these seasonal windows that present more hostile environmental conditions, or that their climatic niche is broader, since they are not restricted to their original distribution as has been proposed in some studies [113]. Of the 30 species of dung beetles found, 17 species are considered possible rare species (<10 individuals), which would suggest that the populations of these beetles are declining [85]. These results agree with Krebs [114], who explained that there were few abundant species and many rare species in the assemblages. Forest remnants contribute to the shelter and conservation of dung beetles; however, conservation strategies must be implemented to preserve and increase the areas of forest fragments and leave biological corridors between forest relicts, as well as agricultural and livestock areas, which preserve dung beetle assemblages.

The dung beetle assemblages, which were composed of members of all three food relocation guilds, were dominated by the paracoprids, which comprised the highest number of species, abundance, and biomass during the rainy season; this was determined by the type of soil (clay and loam-clay), which facilitated the construction of galleries to bury the excrements [115]. The most dominant species among the paracoprids were the small ones, as the large beetles were more sensitive to anthropogenic disturbances and, especially, to the reduction in the number and diversity of wild mammals [116]. The abundance of species in this guild could potentially maintain the functionality of the assemblage despite the absence of large species, provided that their high numbers are maintained [117,118]. The endocoprids, despite having the highest numbers, had lower richness and a biomass that was ten times lower than paracoprids. Telecoprids, which were even less numerous and featured a lower biomass, are considered good bioindicators of anthropogenic disturbances [19,92,119]. Due to its high species number, abundance, and biomass, the paracoprid guild provides essential ecosystem services on cattle farms by recycling the largest amount of cattle manure. Similar results were found by Noriega et al. [61] in three regions of Colombia. In the present study, the higher number of paracoprid species than species of other guilds is similar to the results reported by Barraza et al. [120] and Noriega et al. [88]. Forest fragmentation and deforestation affect the richness and abundance of dung beetles, thus reducing their functional efficiencies, such as nutrient recycling, seed dispersal, and pest control [17]. In livestock agroecosystems, many Scarabaeinae species of the telecoprid guild were disappearing and were replaced by species of the subfamily Aphodiinae. Similar results have been found in mountain forests [121,122].

*4.3. Negative Effects of Livestock Management and Antiparasitic Use*

The effect of Ivermectin on the diversity and structure of dung beetle assemblages showed significant differences in richness, abundance, and biomass between the farms where livestock parasite control did not involve the use of Ivermectin and where a greater diversity and higher numbers of dung beetles belonging to all guilds was observed when compared to those farms where Ivermectin is used routinely. The intensive and long-term use of Ivermectin in cattle farms in the Caribbean region has been detrimental to dung beetle populations. This difference in management to control livestock parasitism has resulted in the loss of 14 species of dung beetles. Different authors found similar results (e.g., [18,33,35,36,40]). For example, in the Czech Republic, Ivermectin-treated sites had ca. 35% lower species richness and 44% lower abundance per dung pat [36]. In SW England, species richness, diversity, and functional diversity were higher on farms with a history of using no parasiticides than on farms with parasiticides. Species of endocoprid (dung dwelling) beetles dominated the community on farms that used parasiticides, particularly

macrocyclic lactones (e.g., Ivermectin), while paracoprid (dung burying) beetles were rare, possibly due to differential impacts depending on life history traits of the functional groups [20].

Ivermectin residues present in the manure of treated cattle are toxic to the dung beetles [123,124] for both larvae and adults [7,31,33,104,125], because they decrease the olfactory and locomotor capacity of the adults [32], thus altering the morphology of the ovaries and stopping vitellogenesis, which causes oocyte resorption and a decrease in fecundity [31,123], which results in the production of fewer larvae [39,126]. This substance can bioaccumulate and affect food chains [124]. As a result, these unintended effects of livestock treatments result in altered structure and diversity of dung beetle assemblages [18,35]. Most dung beetle species are susceptible to Ivermectin [9]. In this study, we found that some native species (*A. crenulatus* and *O. marginicollis*) and the introduced species *D. gazella* were potentially more resistant to Ivermectin use. It is possible that this potential resistance is related to the more saprophagous habits of the Eupariini species. It, therefore, would be important to continue to study *D. gazella* specifically, due to its status as an invasive species in Colombian livestock systems. However, it is also possible that, in the present case, the treatment of cattle did not coincide exactly with the emergence or oviposition period of these species, which appeared relatively unaffected. This situation should be compared to that documented in Mexico with some dung beetle species, whose emergence period coincided with the treatment of pastures with an herbicide [127]. The resilience of dung beetle assemblages to the frequent use of Ivermectin is unknown. Therefore, it is essential to continue studying the spatio-temporal dynamics of the assemblages in wider time windows [35]. At the ecosystem scale, species loss with decreased dung recycling functions was observed after a few years [18].

It is important to highlight that the effects of Ivermectin vary between functional groups [9,36]. The paracoprids and endocoprids were the functional groups who were the most strongly affected by Ivermectin in the Two-IVM 3.15% livestock management. In more general terms, large dung beetles, as well as paracoprid and telecoprid beetles, which are the most functionally efficient in terms of dung removal capacity, are the most vulnerable and the most prone to extinction [35,36,128]. The behavior of these functional groups may be necessary for determining how beetles respond physiologically to Ivermectin [41]. Ivermectin could have differential effects on species, thus affecting larger individuals more dramatically. However, the studies of Verdú et al. [32,124] and Martínez et al. [31] on species of different sizes showed no evident relation between size and vulnerability to ivermectin. The highest risk of extinction of large species is probably more related to their lower reproductive rate (K-strategists) than to lower resistance to Ivermectin. The larger dung beetles would be more exposed to extinction processes than small ones [48]. In our study, we did not find any large species of endocoprids for the three livestock management approaches, and there was only one large species of telecoprids for the No-IVM management (*D. guildingii*) and four large species of paracoprids in the No-IVM management. It is important to mention that there may be a potential bias in the analysis when including the Eupariini, since most of their species have more saprophagous habits; therefore, the abundances in the traps may be affected.

Finally, we did not find any significant preference on the part of the dung beetles for dung from untreated or ivermectin-treated animals (1% and 3.15%). Similar results have been reported by other authors (e.g., [18,42]), but results in the opposite direction have also been reported to show that dung beetles could be attracted to the dung of Ivermectin-treated animals (see [30,33,129,130]). This contradiction is probably only apparent. Ivermectin by itself does not attract dung beetles, but when animals are heavily parasitized, the antiparasitic treatment results in the lysis of the parasites and a strong discharge of several amino acids and volatile organic compounds that change the scent of the dung, which may increase its attractiveness to dung beetles, at least in the few days following the treatment [131].

## 5. Conclusions and Recommendations

In our study, we found that the use of Ivermectin for the control of livestock parasites had negative effects on dung beetle richness, abundance, and biomass in the short term, which can slow down the recycling processes of cattle manure and lead to an accumulation of dung on pastures, thus affecting soil fertility, structure, and aeration, as well as plant species diversity in paddocks, due to non-secondary seed dispersal [132] and less control of biting flies and parasites affecting livestock [133]. For future studies on the effects of Ivermectin on dung beetle assemblages, we suggest monitoring and controlling over time to determine the effects on the soil fauna of the farms. For future management and conservation strategies in the region's livestock systems, the changes in diversity and ecosystem services provided by dung beetles must be considered. Livestock health management has become a major issue, as some veterinary drugs found in dung after treatment of animals can affect the regular interaction of the manure-degrading fauna. In order to reduce the effects of the most toxic molecules on the fauna involved in recycling, the frequency of treatments should be reduced, and the molecules and families of products should be diversified. In addition, older molecules that are still effective and safe for dung beetles should be used, such as benzimidazoles (albendazole) and imidazothiazole derivatives (levamisole) [24]. This would also mitigate the risk of parasite resistance in the case of indiscriminate and continuous use of the same molecule [134,135]. Finally, a simple measure to cut the parasite cycles would be the rotation of cattle, with short times of permanence of the animals in each division.

**Author Contributions:** H.L.T. and J.A.N. conceived the idea and designed the research; H.L.T., B.N. and V.T. gathered the data; H.L.T. and J.A.N. structured the manuscript; H.L.T., J.A.N. and C.M.A.C. analyzed the data and made tables and figures; J.A.N., C.M.A.C. and J.-P.L. corrected and improved different versions of the manuscript, and all authors contributed to the writing and approved the last version of the document. All authors have read and agreed to the published version of the manuscript.

**Funding:** This research received no external funding.

**Data Availability Statement:** Data can be found within the article.

**Acknowledgments:** We thank the owners of the farms involved in this study. We thank Andrés Pérez and Eder Flores for their support in the fieldwork. We thank Jaime Mercado Ordóñez for his knowledge and support in the statistical analysis. We thank José Ramon Verdú for his valuable comments and suggestions to the manuscript. We thank Julián Clavijo Bustos for the taxonomic confirmation of the Aphodiinae and for valuable comments regarding the manuscript. We thank the El Piñal Technical Agricultural Educational Institution in Sucre (Colombia), for providing the necessary time slots to perform this work. PALB was supported by a Master's scholarship from the Minas Gerais State Agency for Research and Development (FAPEMIG).

**Conflicts of Interest:** The authors declare no conflict of interest.

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
