# Peer review of "Effect of Antiparasitic Management of Cattle on the Diversity and Functional Structure of Dung Beetle (Coleoptera: Scarabaeidae) Assemblages in the Colombian Caribbean"

_diversity, doi:10.3390/d15040555_

Round 1

Reviewer 1 Report

This paper analyzes how the use of ivermectin to control internal and external parasites in cattle, could affect the richness, abundance, and biomass of dung beetles in the Colombian Caribbean. The use of this deworming agent is one of the most deeply rooted management practices in cattle ranching and negatively affects the ecological dynamics and ecosystem services provided by different groups of insects, among which dung beetles stand out for their importance in the recycling of cattle manure. As far as I know, data taken in the field are scarce, since the studies carried out on the effect of ivermectin on beetles are more related to their life cycle, body condition, survival and physiology and have been carried out in the laboratory. Another point to consider is that in recent years there has been a decrease in the number of studies that analyze seasonal changes in diversity and in this paper the temporal data of diversity could be related to the dates of application of the deworming agent.

Regarding the text, the paper is easy to read, the authors give sufficient background, clearly explain the problem and objectives of the paper, and include pertinent references. Methods are adequately described and could be replicated in other studies. However, although the design and data analysis are described, there are some factors and concepts not clear and that I would be further explained and considered in all text (see particular comments).

Author Response

Bogotá, 7 March 2023

Dear Dr.

Editor-in-Chief

Diversity

Receive a cordial greeting.

We are resubmitting the manuscript entitled “Effect of antiparasitic management of cattle on the diversity and functional structure of dung beetles (Coleoptera: Scarabaeidae) assemblages in the Colombian Caribbean” (diversity-2195997), by Hernando L. Tovar, Cesar M. A. Correa, Jean Pierre Lumaret, Pablo A. López-Bedoya, Blas Navarro, Valentina Tovar & Jorge Ari Noriega, to be considered for publication as a research article.

At the end of this letter, you will find the original letter with the editor's and reviewers' full comments, together with our replies. We believe these reviews have considerably improved the quality and soundness of the research.

Thank you for considering this manuscript for publication.

Yours sincerely,

Jorge Ari Noriega, on behalf of all co-authors

-----------------------------------------------------------------------------

Reviewer No. 1

General Comments

This paper analyzes how the use of ivermectin to control internal and external parasites in cattle, could affect the richness, abundance, and biomass of dung beetles in the Colombian Caribbean. The use of this deworming agent is one of the most deeply rooted management practices in cattle ranching and negatively affects the ecological dynamics and ecosystem services provided by different groups of insects, among which dung beetles stand out for their importance in the recycling of cattle manure. As far as I know, data taken in the field are scarce, since the studies carried out on the effect of ivermectin on beetles are more related to their life cycle, body condition, survival and physiology and have been carried out in the laboratory. Another point to consider is that in recent years there has been a decrease in the number of studies that analyze seasonal changes in diversity and in this paper the temporal data of diversity could be related to the dates of application of the deworming agent. Regarding the text, the paper is easy to read, the authors give sufficient background, clearly explain the problem and objectives of the paper, and include pertinent references. Methods are adequately described and could be replicated in other studies. However, although the design and data analysis are described, there are some factors and concepts not clear and that I would be further explained and considered in all text (see particular comments). I suggest this article should be published in Diversity, after carrying out the minor changes I have suggested. It is important to have enough data to inform and negotiate with decision-makers on the effect of dewormers on biodiversity.

R./ Thanks to the reviewer for this valuable summary and the compliments about our manuscript. Following your recommendations, we reviewed the document and considered each of your comments.

Particular comments

Introduction

Lines 7-8. You say: the use of agrochemicals (fertilization, pest and weed control), and

the use of veterinary medicinal products in livestock. I suggest: the use of agrochemicals for fertilization, insect control, pest control and the use of veterinary medicinal products in livestock.

R./ Done. We change the sentence.

Line 19. Replace IGAC with Instituto Geográfico Agustín Codazzi

R./ Done.

Line 27. Please insert after (O'Hea et al., 2010) and micro-organisms such as bacteria and fungi.

R./ Done. We insert the sentence.

In the last paragraph of Introduction, you explain that your objectives are related to the following questions: (i) What is the effect of the type of livestock management on dung beetle assemblages (richness, abundance, and biomass) in cattle farms in the Colombian Caribbean? You do not characterize different types of cattle management and compare them, you are analyzing only one variable which is the control of internal and external parasites using Ivermectins at two different doses and not using it. Then it should be better: What is the effect of use of Ivermectin on dung beetle assemblages (richness, abundance, and biomass) in cattle farms in the Colombian Caribbean?

R./ Done. We change the objective.

(ii) Do the functional groups present different responses for types of livestock management?

It should be better: Do the functional groups of dung beetles present different responses for use of Ivermectin?

R./ Done. We change the objective.

(iii) Are there species associated with each feces type (e.g., dung without IVM, with a low dose (IVM 1%), and with a high dose (IVM 3.15%)?

You only used cow manure from animals that were treated or not with ivermectin. You do not use different feces types (sheep, horse, donkey, dog, etc).

Please, consider change type of livestock management with use of Ivermectin in them entire document. Remember if you only considered bovines in the study, use cattle management or ranching management, because livestock include all type of animals.

R./ Done. We agree with the reviewer, and we made this change.

Material and methods

Study area

Lines 7-8. Add dual-purpose farming if applicable: Cattle are usually managed in extensively used pastures to produce milk and meat (dual-purpose farming)

R./ Done. We add the clarification.

Livestock management

Lines 1-3. You say: In order to assess and compare the effect of the type of livestock management (use of Ivermectin, IVM) on the diversity and structure of dung beetle assemblages, the study was carried out on nine livestock farms in the Colombian Caribbean, at least 3 to 5 km apart from each other.

I suggest: To assess and compare the effect of the use of Ivermectin (IVM) on the diversity and structure of dung beetle assemblages, the study was carried out on nine livestock farms in the Colombian Caribbean, at least 3 to 5 km apart from each other.

R./ Done. We change the sentence.

Lines 4-9. You say: The selected farms represented the typical animal production system of the Colombian Caribbean region, with three types of parasite control: i) NO-IVM = Ivermectin was not applied (nor any other product for more than 30 years), ii) TWO-IVM 1% = Two doses of Ivermectin at 1% concentration per year, and iii) TWO-IVM 3.15% = Two doses of Ivermectin at 3.15% concentration per year. These two doses were usually applied, one in February (dry season) and the other in October (rainy season).

I suggest: Selected farms represent the typical parasite control in the Colombian Caribbean region: two doses of Ivermectin at 1% or 3.5% concentration, usually applied twice each year: one in February (dry season) and the other in October (rainy season). In some farms there are not control of parasites. It is not necessary include Two in the abbreviatures if you detail this procedure.

R./ Done. We understand the reviewer's point of view, and we did some of the proposed changes. However, we keep the conventions that we use to avoid changes in all figures and tables.

Sampling and experimental design

Last lines in the first paragraph: I suggest change controlled with not controlled, because it seems that treated and controlled animals are similar treatments.

R./ Done. We change it.

Second paragraph, lines 3-6. The experimental design included three factors: "Livestock management" (NO-IVM, TWO-IVM 1%, and TWO-IVM 3.15%), "Season" (dry and rainy), and "Treatments in the baits" (without IVM, IVM 1 %, and IVM 3.15%). 3

Please, could you explain better your experimental design? The difference between the first and third factors is not clear. Your intention is to separate the effect due directly to the animal (Ivermectin treatment in cows) and the effect coming from the treated cattle manure?

R./ Done. We explain the factors in a more detailed way.

Third paragraph, line 6. Why 4 days later? although you mention a bibliographic reference, it would be better to explain it.

R./ We add a more detailed explanation to the selection of four days.

Dung beetles' diversity, identification, and biomass

First paragraph, last lines: Please, use upper- and lower-case letters in OHAUS CORPORATION, USA.

R./ Done.

Data analysis

The authors use two different concepts related to manure relocation, guilds, and functional groups in text and figures (burrowers, rollers, and dwellers, or paracoprids, endocoprids, telecoprids). It is necessary to explain the reasons because cause confusion in readers. Please, consider the same concept in the entire paper and consult the following studies:

Tonelli, M. (2021). Some considerations on the terminology applied to dung beetle functional groups. Ecol. Entomol. 46, 772-776.

Fauth, J. E., Bernardo, J., Camara, M., Resetarits Jr, W. J., Van Buskirk, J., & McCollum, S. A. (1996). Simplifying the jargon of community ecology: a conceptual approach. The American Naturalist, 147(2), 282-286.

R./ Done. Thanks to the reviewer for pointing out this issue. We check the whole document and unify the terminology related to guilds and functional groups.

Differences in dung beetle assemblages between livestock treatments

Please change livestock treatments and/or livestock management with dewormer treatment or Ivermectin treatments, as appropriate.

R./ Done. We change it.

Differences in dung beetle relocation guilds between livestock treatments

Please change livestock treatments and/or livestock management with dewormer treatment or Ivermectin treatments, as appropriate. Maintain the terminology you selected in terms of functional groups, guilds, and food re-localization.

R./ Done. We change it.

Discusion

Dung beetles’ assemblage structure according to seasonality

Although the results are compared between seasons and the arguments in Discussion include environmental variables, beetle physiology and human activities; I suggest including more reflections about the relation between deworming application in February and October and changes in the temporal dung beetle diversity in your paper.

R./ Thanks to the reviewer for pointing out this issue. However, we checked all the available literature and did not find any article that discusses experimentally the seasonal effect related to Ivermectin use. This is a topic that needs more studies.

Conclusion

Please finish the section with this comment: Finally, a simple measure to cut the parasite cycles is the rotation of cattle, with short times of permanence of the animals in each division.

R./ Done. We include this sentence.

Tables and Figures

Consider in the legends of figures and tables the changes carry out in all text and use the same font size and type (v.g. In the figures or tables where food relocation, guilds or functional groups are included, use the selected terms).

R./ Done. We standardize the functional groups terms. We check Tables and Figures and unify font type. However, font size is impossible to unify because of the amount of information and the structure of the tables. 

In the legend of Table 2, change paracopridos with paracoprids. The size of font in this Table is smaller than in others, please use the same size of font in all Tables 4

R./ Done. We corrected the typo. Font size is impossible to unify because of the amount of information and the table's structure.

Figure 3. I suggest: Venn diagram showing proportion of shared dung beetle species among the assemblages sampled in cattle farms with different Ivermectin treatments.

R./ Done. We change it.

Figure 5. If you show the differences among functional groups, use the same scale for individuals, the same scale for species richness and the same scale for biomass in the three treatments. For example is not the same 4000 individuals or 3000 individuals than 100. The idea of the differences is not real.

R./ We understand the point of view of the reviewer, but in this case, the scales are so different in terms of magnitude that if you unify scales is going to be impossible to show the statistical difference inside the treatments. In a quick look, it is possible that the reader does not see the difference, but if you check the scales, it is easy to understand that the magnitude of the values is different.  

Figure 6. Please explain the meaning of the arrows and the direction of them.

R./ Done. We explain it in detail in the text.

References in text

Please, use the same style in all references, in some times you use coma and in others you do not (v.g. FEDEGAN, 2018 vs ICA 2022).

R./ Done. We checked all the references and standardized this.

References list

All references are centered in the pages, please change them to left.

R./ Done.

Check carefully that all references in the text are listed in the reference list.

R./ Done.

-------------------------------------------------------------------------------

Reviewer No. 2

This hyphotesis is hard to test given the work was performed for only one year; that is, a single replication through time. Recommend to rewrite as a short-time hyphotesis.

R./ Done. We check the hypothesis and re-wrote it.

Specify this refers to vegetation type

R./ Done. We specify the reference.

What about the effect of farms? Between farm variability may add variance to the results due to climatic, zonal differences.

R./ We understand the point of view of the reviewer. However, there are no substantial climatic or zonal differences between the farms because all of them are located in the same region. In addition, we try to select very similar farms in terms of size, vegetation structure, use, and management history to reduce this potential effect.

Describe test types applied to comparisons

R./ We used GLMMs to analyze and compare the effects of livestock management on the abundance, species richness, and biomass of dung beetles. We mentioned this in the text.

Did you define this in M&M?

R./ We do not understand the comment of the reviewer.

Kind of confuse, did or did not show differences? p < 0.01 indicates difference

R./ Done. Thanks to the reviewer for pointing out this issue. We check and correct these sentences.

This whole paragraph is not ddirectly linked to the work because no comparison was performed between un-altered Bs-T and transformed biomes to cattle production systems

R./ We understand the point of view of the reviewer. However, we believe it is essential to add to the analysis a regional context regarding the loss of vegetation cover. All the pastures that we included in our study were dry forests that suffered a clearcut.

It is the reverse, urban settlements impact Bs-T biomes, it should be its vulnerability or the like

R./ Done. We change it.

It is highly speculative given to sampling was performed on adyacent forest land

R./ We understand the point of view of the reviewer. However, we know that the presence of forest fragments in this region has a positive effect on the assemblage structure. Pastures located near a forest fragment have a richer assemblage than pastures far from these vegetation relicts. In addition, dung beetle sampling was done directly on pastures at a high distance from the forests.     

How does it compares with the catched species in this work?

R./ Done. Thanks to the reviewer for pointing out this idea. We incorporate this idea into the text.

Cannot claim this conclusion, the paper did not address comparison in original and tranformed vegetation types.

R./ We understand the point of view of the reviewer. However, we disagree. Our data show that large beetles disappear when the amount and intensity of Ivermectin increases.

It is more of recommendations than conclussions about what was obtained in the research. Should address main findings

R./ Following the reviewer's suggestion, we add the term “recommendations” to the subtitle.

Check statement, look more like boxplots, which provide with quantile values

R./ Done. We checked the statement and corrected it.

Check as in figure 4

R./ Done. We checked the statement and corrected it.

Show X axis values, title. What non-tagged symbols mean?

R./ This type of figure does not have an x-axis, so we add a convention level in the upper section of the graph. The rest of the circles represent the other species in each treatment. Probably, the reviewer does not know this type of graph, which is widely used in this type of study.

Looks more like a boxplot, if so, extreme values does not represent SEM. What is the  value inside the box? What does the box borders mean? What test was applied to compare treatments? Describe tests in M&M. Twice a SE comprise approximately 95%, thus, the overlap should indicate no differences in mean values if interpreted as in text, explain

R./ Done. We checked the statement and corrected it.

Reviewer 2 Report

Extensively detailed review of the literature, excellent account. Well designed methods for meeting experimental objectives. Results and discussion present extensive, useful details of the study. Overall, well written, no edits to language needed. Figures effectively convey key results. 

Author Response

(The authors gave the same response as above.)

Reviewer 3 Report

General comment

The paper provides just another example of how ivermectin affects coprophagous beetles. While original in location, it does not provide much insight into what is know about how ivermectin influence dung beetles. It is clear and well written. Title, hyphotesis, discussion and conclussions tend to inflate the reachings of the paper. Thus it is recommended the authors delineate limitations of the work. For example, hypothesis refers to long term effects, but the research is just one year, a single replication for seasons. Discussion involved comparison with un-altered surrounded vegetation, however the paper did not sample places other than cattle ranchs. Conclussions include recommendations on management but management alternatives to minimize negative effects of ivermectin were not evaluated.

Title

What the paper really address is studying the effect of ivermectin on dung beetles in Colombian cattle ranchs, thus I suggest changing the title to include this feature.

Abstract

Relate information to the findings in the experimental work, not linked to original Bs-T regions

MM

Include tests performed.

Discussion

Initial discussion paragraph, as well as the whole section 4.1 is not directly related to the work because no  comparison was performed between altered and un-transformed dry forest in the current work.

Much of the discussion address a comparison between altered and untransformed forest coverage, which was not included in the current research. Thus, the discussion is speculative. Discussion should center in comparisons related to the treatments within an altered forest coverage.

References in the text should include 3 papers max for a given discussion subject.

Figures

Check and clarify if figures 4, 5 are really means and SEM, more likely they are boxplots, representing qunaatiles measures of the samples.

Additional comments in the attached file.

Author Response

(The authors gave the same response as above.)
